# Effects of HIV Infection in Plasma Free Fatty Acid Profiles among People with Non-Alcoholic Fatty Liver Disease

**DOI:** 10.3390/jcm11133842

**Published:** 2022-07-02

**Authors:** Javier Martínez-Sanz, María Visitación Calvo, Sergio Serrano-Villar, María Luisa Montes, Rosa Martín-Mateos, Diego Burgos-Santamaría, Jorge Díaz-Álvarez, Alba Talavera-Rodríguez, Marta Rosas, Santiago Moreno, Javier Fontecha, Matilde Sánchez-Conde

**Affiliations:** 1Department of Infectious Diseases, Hospital Universitario Ramón y Cajal, Instituto Ramón y Cajal de Investigación Sanitaria (IRYCIS), 28034 Madrid, Spain; serranovillar@gmail.com (S.S.-V.); jolohinodam@gmail.com (J.D.-Á.); mrcancio.3@gmail.com (M.R.); smguillen@salud.madrid.org (S.M.); 2CIBER de Enfermedades Infecciosas (CIBERINFEC), Instituto de Salud Carlos III, 28029 Madrid, Spain; 3Food Lipid Biomarkers and Health Group, Institute of Food Science Research (CIAL, CSIC-UAM), 28049 Madrid, Spain; mv.calvo@csic.es (M.V.C.); j.fontecha@csic.es (J.F.); 4HIV Unit—Internal Medicine Service, Hospital Universitario la Paz—IdiPAZ, 28046 Madrid, Spain; mmontesr2001@yahoo.es; 5Department of Gastroenterology and Hepatology, Metabolic Liver Disease Clinic, Hospital Universitario Ramón y Cajal, Instituto Ramón y Cajal de Investigación Sanitaria (IRYCIS), 28034 Madrid, Spain; rosamartinmat@hotmail.com (R.M.-M.); diegoburgossantamaria@gmail.com (D.B.-S.); 6CIBER de Enfermedades Hepáticas y Digestivas (CIBEREHD), Instituto de Salud Carlos III, Universidad de Alcalá, 28871 Madrid, Spain; 7Bioinformatics Unit, Hospital Universitario Ramón y Cajal, Instituto Ramón y Cajal de Investigación Sanitaria (IRYCIS), 28034 Madrid, Spain; albatalavera92@gmail.com

**Keywords:** non-alcoholic fatty liver disease, HIV, lipidomics, fatty acids

## Abstract

Despite its high prevalence, the mechanisms underlying non-alcoholic fatty liver disease (NAFLD) in people living with HIV (PLWH) are still unclear. In this prospective cohort study, we aim to evaluate differences in plasma fatty acid profiles between HIV-infected and HIV-uninfected participants with NAFLD. We included participants diagnosed with NAFLD, both HIV-infected and HIV-uninfected. Fatty acid methyl esters were measured from plasma samples. Ratios ([product]/[substrate]) were used to estimate desaturases and elongases activity. We used linear regression for adjusted analyses. We included 31 PLWH and 22 HIV-uninfected controls. We did not find differences in the sum of different types of FA or in FA with a greater presence of plasma. However, there were significant differences in the distribution of some FA, with higher concentrations of ALA, *trans*-palmitoleic, and behenic acids, and a lower concentration of lignoceric acid in PLWH. PLWH had lower C24:0/C22:0 and C16:0/C14:0 ratios, which estimates the activity of elongases ELOVL1 and ELOVL6. Both groups had similar fatty acid distribution, despite differences in traditional risk factors. PLWH had a lower proportion of specific ratios that estimate ELOVL1 and ELOVL6 activity, which had been previously described for other inflammatory conditions, such as psoriasis.

## 1. Introduction

The prevalence and morbidity of non-alcoholic fatty liver disease (NAFLD), recently redefined as metabolic dysfunction-associated fatty liver disease (MAFLD) [1], is increasing in people living with HIV (PLWH), and is a leading cause of end-stage liver complications in this population [2]. Despite its high prevalence, mechanisms underlying NAFLD are being explored [3]. Different factors have been proposed to explain the higher risk of NAFLD in PLWH, including antiretroviral therapy, past exposure to hepatotoxic drugs, such as stavudine and didanosine, persistent immune activation, chronic inflammation, and microbiome alterations. In addition, PLWH have a higher prevalence of traditional risk factors, such as insulin resistance and visceral adiposity [3].

Dietary fatty acids (FA) play an important role in the development of NAFLD [4]. The lipidome composition may compromise hepatic lipid metabolism, with different FA involved in the pathogenesis of liver disease, which may be related to the development and prevention of NAFLD [5]. Bioactive FA are involved in many other metabolic pathways, including the regulation of inflammatory processes [6]. Alterations in the composition of FA relate to inflammation in the HIV-uninfected population [7]. In PLWH, altered metabolic profiles are connected to the virus itself, its treatment, and persistent inflammation [8]. Plasma lipidome is altered in PLWH under suppressive antiretroviral therapy, with a relationship found between some lipid species and elevated cardiovascular risk in this population [9]. Different factors associated with increased incidence of NAFLD in PLWH have been postulated [2]. Still, to date, there are no studies that evaluate the differences in lipidome for both those with and without HIV infection. Previous work studied lipidome and elevated liver enzymes in HIV-infected and HIV-uninfected subjects without considering NAFLD diagnosis, yielding mixed results [10]. Our objective is to assess differences in plasma FA profiles between HIV-infected and HIV-uninfected people with NAFLD.

## 2. Materials and Methods

### 2.1. Study Design and Population

This was a prospective cohort study at Hospital Universitario Ramón y Cajal in Madrid (Spain) from January 2018 to December 2018. HIV-uninfected participants diagnosed with NAFLD were recruited at the Metabolic Liver Disease Clinic. HIV-infected participants receiving suppressive antiretroviral therapy were recruited at the HIV Clinic. We included those who presented elevated liver enzymes for at least two determinations, separated by six months. Any transaminase (GGT, ALT, AST) level above the upper limit of normal in our laboratory was considered. HIV-infected participants were receiving antiretroviral therapy for at least 1 year with undetectable viremia in the last 6 months. All participants underwent abdominal ultrasound and a screening analysis for liver disease. Diagnosis of NAFLD was established by ultrasound confirmation of steatosis and exclusion of other aetiologies of chronic liver disease. Exclusion criteria included viral hepatitis; alcohol abuse (defined by >30 g daily in men and >20 g daily in women); cocaine, heroin, or designer drug abuse; other known liver diseases (autoimmune, genetic, drug-related); isolated alkaline phosphatase alteration; recent drug toxicity; impossibility of cannulating a peripheral blood line if a liver biopsy is required; pregnancy or desired pregnancy; decompensated liver disease or hepatocarcinoma; and any other comorbidity that, at the investigator’s discretion, could prevent correct compliance with the study protocol. Clinical information was collected through the clinical interview at the baseline visit of the study, together with the review of the medical history recorded in the electronic medical record. Metabolic syndrome was defined according to the National Cholesterol Education Program (NCEP) third report [11]. The study was approved by the Institutional Review Boards of the Carlos III Health Institute, Madrid, Spain (Project PI 17/01717), and by the Ethics Committee at the University Hospital Ramón y Cajal (ceic.hrc@salud.madrid.org, Approval Number 097/17). All patients gave written informed consent before initiation of the study procedures. All methods were performed in accordance with the relevant guidelines and regulations.

### 2.2. Determination of Fatty Acid Methyl Esters in Plasma and Serum

Fatty acid methyl esters (FAME) were prepared directly from plasma and serum samples (200 μL), without previous lipid extraction, following the acid-base methylation method as described by Castro-Gomez et al. [12]. Two independent methylation processes were carried out for each sample. Tritridecanoin (Sigma-Aldrich, Burlington, MA, USA) was added as an internal standard (75 μL; 1.0 mg/mL) to samples prior to methylation.

FAME analysis was carried out on an Autosystem chromatograph (Perkin Elmer, Beaconsfield, UK), fitted with a flame ionization detector (FID). Helium was the carrier gas with a column inlet pressure, set at 20 psi and a split ratio of 1:20. The injection volume was 0.5 μL of the sample. The column, a VF-23ms-fused silica capillary column (30 m × 0.25 mm i.d. × 0.25 μm film thickness: Varian, Middelburg, The Netherlands) was held at 60 °C for 1 min after injection. Temperature-programmed at 10 °C/min to 130 °C, then at 3 °C/min to 170 °C, and finally at 10 °C/min to 230 °C, where it was held for 5 min. The injector and detector temperatures were set at 250 °C and 270 °C, respectively. For FAME determination and quantification, anhydrous milk fat (reference material from Fedelco Inc., Madrid, Spain) and FA standards (Larodan, Barcelona, Spain) were used. FAME levels in plasma and serum were compared without significant measurement differences; the present work shows comparisons of plasma samples. FA concentrations were expressed as molecular percentages (g FA/100 g fat).

### 2.3. Statistical Analysis

Baseline characteristics of patients with and without HIV infection were compared with the Wilcoxon rank-sum test for continuous variables and Fisher’s exact test for categorical variables. We used ratios ([product]/[substrate]) as indirect estimators of desaturases and elongases’ activity, in accord with previous literature [13]. We performed a principal component analysis (PCA) and used a biplot to visually inspect the data matrix, projecting different FA in two-dimensional graphical form, as per HIV status. Differences between individual FA and ratios were compared with linear regression, including the baseline characteristics that showed significant differences between the two groups as covariates. Since the objective of this work was exploratory, with a rational selection of the ratios to be evaluated (based on the background literature), we did not use corrections to adjust for multiple comparisons [14]. Analyses were performed with Stata v. 17.0 (StataCorp LP, College Station, TX, USA). Figures were created using GraphPad Prism v9.2 (GraphPad, La Jolla, CA, USA).

## 3. Results

We included 53 participants diagnosed with NAFLD, of whom 31 (58%) were PLWH and 22 (42%) were HIV-uninfected (Figure 1). Participant demographic information is provided in Table 1. In summary, participants with HIV were younger, with a higher percentage of males, lower BMI, and a lower percentage of diabetes mellitus (DM) and metabolic syndrome. The severity of NAFLD was comparable in both groups, as was diet, except for lower dairy consumption and higher alcohol consumption in PLWH. Based on these results, to avoid overfitting and collinearity, subsequent analyses were adjusted for sex, BMI, and alcohol and dairy consumption. BMI was used as a variable related to the prevalence of both DM and metabolic syndrome. However, additional multivariate models were fitted, adjusting for the presence of DM or metabolic syndrome, with similar estimates. At inclusion, all HIV-infected participants were receiving antiretroviral therapy (94% under an integrase-strand inhibitor-based regimen) for an average of 6 years while maintaining a suppressed viral load. Median CD4+ T-cell count was 721 (IQR 581–981) cells/μL and the median CD4/CD8 ratio was 0.81 (IQR 0.58–1.29).

Figure 2 shows the plasma FA concentration in both groups. Palmitic, oleic, linoleic, stearic, and arachidonic were the primary FA, with a similar distribution in both groups. The adjusted differences between groups in all individual FA concentrations are shown in Appendix A. Appendix A reveals the biplot resulting from PCA. The explained level of variance is 27%, and both groups appear to be overlapping; as such, HIV infection does not significantly influence the composition of FA. There was no difference in overall analysis in any FA group, including saturated (SFA), monounsaturated (MUFA), polyunsaturated (PUFA), omega-3, and omega-6 FA. In individual comparisons, PLWH had a higher concentration of α-Linolenic (ALA), *trans*-palmitoleic (*trans* C16:1), and behenic (C22:0) acids, and a lower concentration of lignoceric (C24:0) acid. Moreover, they had a marginally significant higher concentration of myristic (C14:0) and pentadecanoic (C15:0) acids. These results are seen graphically in Figure 3. Appendix A shows different ratios for the estimated activity of desaturases and elongases. PLWH have a lower C24:0/C22:0 ratio, which gauges the activity of elongase ELOVL1, and a lower C16:0/C14:0 ratio, which estimates the activity of ELOVL6 (Figure 3).

## 4. Discussion

This prospective study found a similar plasma FA profile in people diagnosed with NAFLD, regardless of HIV status. We did not find differences in the sum of different types of FA or in FA with a greater presence of plasma. However, there were significant differences in the distribution of some FA, with higher concentrations of ALA, *trans*-palmitoleic, and behenic acids and a lower concentration of lignoceric acid in PLWH.

Our findings agree with previous work studying plasma FA profiles in patients with NAFLD [15,16]. All primary FA in our study had been previously associated with different metabolic indicators, e.g., body fat, LDL cholesterol, and liver enzymes [15]. In addition, concentrations of major FA were similar to those previously reported with NAFLD but different in healthy controls [16]. To the best of our knowledge, no previously published data exists for different FA profiles in PLWH with NAFLD. This is the first study to explore it: consequently, results cannot be compared to the previous literature.

It is postulated that PLWH are at increased risk for NAFLD due to numerous factors associated with HIV infection. These include sustained immune activation and inflammation, and toxicity of antiretroviral therapy, especially for nucleoside reverse transcriptase inhibitors (NRTIs) used in the past. Other antiretroviral drugs, such as efavirenz, also present an increased risk of chronic liver disease [17], while modern non-nucleoside reverse transcriptase inhibitors (e.g., rilpivirine) appear to improve liver fibrosis [18]. Moreover, traditional risk factors are more prevalent in this population, including toxic intake, unhealthy diet, and a sedentary lifestyle [2]. HIV infection itself, and related factors, are associated with a higher prevalence of dyslipidaemia in PLWH [8]. In sample participants with NAFLD, we observed that PLWH have similar steatosis severity, being younger with lower BMIs and a lower prevalence of metabolic syndrome, the same cholesterol levels, and use of lipid-lowering drugs. This is seen in context of an increased risk due to factors other than metabolic syndrome, justifying the study of FA profiles in these different populations.

Strikingly, despite a lower prevalence of risk factors, PLWH have a plasma FA profile comparable to non-HIV NAFLD controls. As mentioned, however, some significant differences were observed, especially lower C24:0/C22:0 and C16:0/C14:0 ratios, which estimate the activity of the elongases ELOVL1 and ELOVL6, respectively [19,20]. Elongases belong to the *elongation of very long chain FA* (ELOVL) microsomal enzyme family, which regulates the elongation of SFA and MUFA. Both ELOVL1 and ELOVL6 are biomarkers associated with an increased risk of NAFLD in the HIV-negative population with metabolic syndrome [20,21]. These enzymes, responsible for the last steps in the synthesis of SFA, control FA tissue composition. It has been proposed that inhibition of ELOVL6 activity could be a new therapeutic approach to combat insulin resistance, metabolic syndrome, and cardiovascular risk [22]. Although there are no data in PLWH, previous work found decreased ELOVL1 activity in other models of chronic inflammation, such as psoriasis [23]. It was found that interferon-γ (which plays an important role in HIV infection) produces downregulation of ELOVL, decreasing ceramides with long-chain FA, in turn, linked to the development of NAFLD in obese individuals with metabolic syndrome [23].

Our study has limitations. FA distribution in plasma has more variability than that studied in erythrocyte membranes. However, data may still be compared: the characteristics of dietary patterns were similar in both groups, and we performed adjusted analyses. The groups showed differences in variables such as sex, BMI, diabetes mellitus, and metabolic syndrome. Statistical models were adjusted for sex and BMI. We used ratios to estimate elongases activity but did not measure enzyme expression. This method has been widely used in the literature, so we consider the estimates to be valid. However, the exploratory nature of the data and the restricted sample size demands caution when interpreting the results.

## 5. Conclusions

We found that although PLWH present a different classical risk factor profile than HIV-uninfected individuals with NAFLD, the plasma FA profile is similar in both groups. However, we found differences that may be justified by the immunological characteristics of the HIV-infected population. Given the burden of NAFLD in PLWH, further studies are warranted to both advance knowledge and to control this disease.

## Figures and Tables

**Figure 1 jcm-11-03842-f001:**
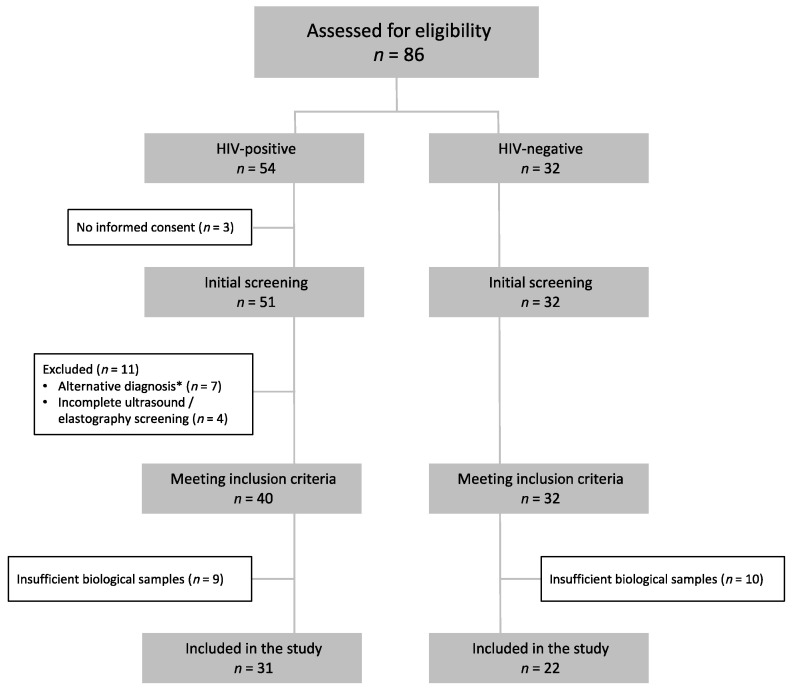
Study flowchart. Participants were assessed for eligibility if they met the study’s analytical inclusion criteria and did not have a diagnosed alternative liver disease. * Alterative diagnoses were: Hepatitis C virus infection (*n* = 5), autoimmune hepatitis (*n* = 1), primary biliary cholangitis (*n* = 1).

**Figure 2 jcm-11-03842-f002:**
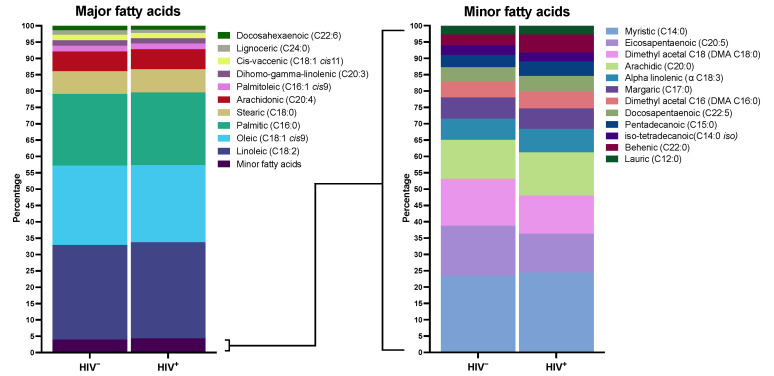
Distribution of fatty acid methyl esters in plasma. This figure shows plasma fatty acid distribution in participants with NAFLD and HIV infection and controls without it. The left panel shows major plasma fatty acid distribution in both groups. The right panel shows fatty acid distribution with lower plasma representation. Data are expressed as percentages, with a total sum of 100% in each population.

**Figure 3 jcm-11-03842-f003:**
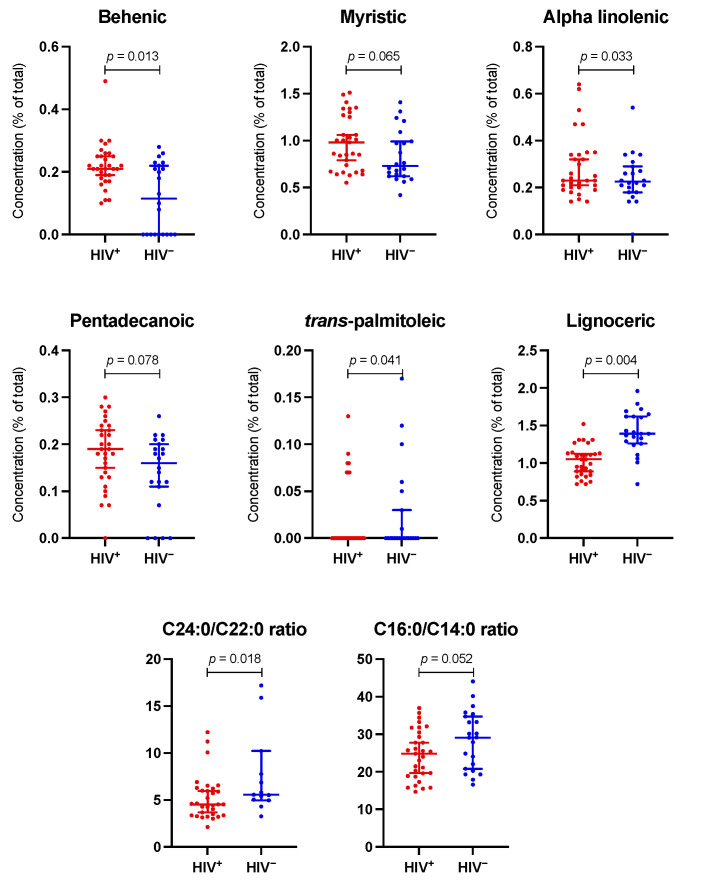
Main differences in plasma fatty acids and estimated activity of ELOVL1 and ELOVL6 elongases, according to HIV status. The dot plots show the most significant differences between the two groups regarding fatty acids plasma concentrations and calculated ratios. C24:0/C22:0 and C16:0/C14:0 ratios estimate the activity of ELOVL1 and ELOVL6 elongases, respectively. The *p*-value is obtained from the adjusted linear regression model.

**Table 1 jcm-11-03842-t001:** Population baseline characteristics according to HIV status.

	HIV Positive (*n* = 31)	HIV Negative (*n* = 22)	*p*-Value
**Age, median (IQR)**	56 (46, 61)	60 (58, 71)	0.005
**Sex, *n* (%)**			
Male	27 (87)	9 (40)	0.001
Female	4 (13)	13 (59)	
**NAFLD severity**			
NAFLD-LFS, median (IQR)	−0.05 (−1.13, 3.57)	1.78 (0.29, 2.62)	0.293
Controlled attenuation parameter (CAP), median (IQR)	275 (234, 288)	313 (273, 357)	0.053
**Body mass index, median (IQR)**	27 (25, 28)	33 (31, 36)	<0.001
**Hypertension, *n* (%)**	10 (32)	13 (59)	0.091
**Diabetes mellitus, *n* (%)**	4 (13)	10 (45)	0.006
**Metabolic syndrome, *n* (%)**	6 (19)	17 (77)	<0.001
**Ischemic heart disease, *n* (%)**	0 (0)	2 (9)	0.158
**Stroke, *n* (%)**	0 (0)	1 (5)	0.415
**Tobacco use, *n* (%)**	5 (16)	2 (9)	0.885
**Illicit drug use, *n* (%)**	2 (6)	0 (0)	0.112
**Lipid-lowering agents, *n* (%)**			
Statins	13 (42)	11 (50)	0.871
Ezetimibe	2 (6)	2 (9)	0.590
Fibrates	3 (10)	1 (5)	0.633
**Diet (food group, servings per week), median (IQR)**			
Legumes	2 (1, 3)	2 (1, 2)	0.249
Cereals	7 (7, 7)	7 (4, 7)	0.860
Vegetables	5 (4, 7)	6 (4, 8)	0.386
White fish	1 (0, 2)	1 (1, 3)	0.221
Blue fish	1 (1, 3)	1 (1, 2)	0.986
Red meat (beef)	2 (1, 3)	1 (1, 2)	0.116
Pork	1 (1, 2)	2 (1, 2)	1.000
Poultry	3 (2, 4)	3 (2, 5)	0.811
Dairy products	13 (7, 14)	14 (11, 14)	0.056
Fats (oil, butter)	7 (7, 7)	7 (5, 7)	0.809
Alcohol	1 (3, 7)	0 (0, 0)	0.003
Coffee or tea	7 (7, 7)	7 (7, 7)	0.937
Soft drinks	2 (0, 3)	0 (0, 2)	0.136
**Liver enzymes, median (IQR)**			
GGT (U/L)	63 (47, 106)	60 (37, 125)	0.787
ALT (U/L)	35 (30, 49)	48 (33, 63)	0.299
AST (U/L)	28 (24, 39)	34 (27, 43)	0.338
Total bilirubin (mg/dL)	0.7 (0.6. 0.9)	0.7 (0.5, 0.9)	0.709
**Lipid profile, median (IQR)**			
Total cholesterol (mg/dL)	202 (178, 215)	194 (156, 223)	0.658
LDL (mg/dL)	116 (92, 144)	104 (79, 147)	0.714
HDL (mg/dL)	44 (35, 54)	47 (41, 48)	0.510
Triglycerides (mg/dL)	168 (108, 224)	118 (84, 148)	0.059

Abbreviations: ALT, alanine aminotransferase; AST, aspartate aminotransferase, GGT, gamma-glutamyl transferase; IQR, interquartile range; HDL, high-density lipoprotein; LDL, low-density lipoprotein, NAFLD-LFS, non-alcoholic fatty liver disease-liver fat score.

## Data Availability

Data available on request. The data presented in this study are available on request from the corresponding author.

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
