# Peer review of "Effects of HIV Infection in Plasma Free Fatty Acid Profiles among People with Non-Alcoholic Fatty Liver Disease"

_jcm, 2022, doi:10.3390/jcm11133842_

Round 1

Reviewer 1 Report

The manuscript “Effects of HIV infection on plasma free fatty acid profiles among people with nonalcoholic fatty liver disease” is interesting and of significant importance in its field. However, the description of the methods does not allow us to understand how the study was developed due to the lack of information on how the data were collected. Therefore, a major revision of the text is necessary for a better understanding of the study:

1- Describe in methods the details the inclusion and exclusion criteria, and procedures. Remove these items from the supplementary material

2- Include in the methods how the data described in Table 1 were obtained - diabetes, hypertension, metabolic syndrome, lipid-lowering agents use, BMI and food intake. What is the source of the information, medical records or interview?

3- Include de Figure S1. Study flowchart in the item results. Remove in the supplementary material.

4- Regarding the lipid profile, the data was collected from the medical record, what was the deadline considered for this survey, weeks or months?

5- Include in the study's limitations paragraph that the groups are different in terms of gender, BMI, Diabetes mellitus and metabolic syndrome.

Author Response

Manuscript ID: jcm-1784611

Article title: Effects of HIV infection in plasma free fatty acid profiles among people with non-alcoholic fatty liver disease.

The authors would like to thank the Reviewers and Editor for their careful review of our manuscript and their constructive comments and suggestions for improving its quality. The following responses have been prepared to address the referees’ comments point-by-point. Line numbers refer to the revised manuscript with tracked changes.

REVIEWER #1 COMMENTS:

The manuscript “Effects of HIV infection on plasma free fatty acid profiles among people with nonalcoholic fatty liver disease” is interesting and of significant importance in its field.

Response: The authors would like to thank the Reviewer for the positive assessment of our manuscript and for recognizing the importance of the subject studied.

However, the description of the methods does not allow us to understand how the study was developed due to the lack of information on how the data were collected. Therefore, a major revision of the text is necessary for a better understanding of the study:

1- Describe in methods the details the inclusion and exclusion criteria, and procedures. Remove these items from the supplementary material.

Response: Following the Reviewer’s suggestion, we have included the eligibility criteria and procedures in the Methods section (lines 68-81), and we have removed these items from the Supplementary Material.

2- Include in the methods how the data described in Table 1 were obtained - diabetes, hypertension, metabolic syndrome, lipid-lowering agents use, BMI and food intake. What is the source of the information, medical records or interview?

Response: Clinical data were collected through the clinical interview at the baseline visit of the study, together with the review of the medical history recorded in the electronic medical record. Following the Reviewer's request, this information has been included in the manuscript (lines 82-84)

3- Include de Figure S1. Study flowchart in the item results. Remove in the supplementary material.

Response: Following the Reviewer’s suggestion, we have included the study flowchart in the Results section (Figure 1), and we have removed this item from the Supplementary Material.

4- Regarding the lipid profile, the data was collected from the medical record, what was the deadline considered for this survey, weeks or months?

Response: The lipid profile data were collected in the blood test performed at the baseline visit of the study, or through medical records up to 3 months earlier.

5- Include in the study's limitations paragraph that the groups are different in terms of gender, BMI, Diabetes mellitus, and metabolic syndrome.

Response: Following the Reviewer’s suggestion, we have included this limitation (lines 289-290).

Reviewer 2 Report

The authors present an interesting paper about differences between HIV infected and HIV uninfected patients regarding the lipid profiles in the setting of non-alcoholic fatty liver disease (NAFLD). The manuscript is well written and the results and discussion are well explained however some minor issues should be addressed:

  • Metabolic syndrome is present in both HIV-infected and uninfected patients, specially among the latter. However, I could not find how metabolic syndrome was defined. Is it defined according to the Executive Summary of the Third Report of the National Cholesterol Education Program (NCEP) (Expert Panel on Detection, Evaluation, and Treatment of High Blood Cholesterol in Adults. Executive Summary of The Third Report of The National Cholesterol Education Program (NCEP) Expert Panel on Detection, Evaluation, And Treatment of High Blood Cholesterol In Adults (Adult Treatment Panel III). JAMA. 2001 May 16;285(19):2486-97. doi: 10.1001/jama.285.19.2486. PMID: 11368702) or another standard?

  • Table 1 shows the most common commorbidities within the study population. However, no mention to cardiovascular disease (i.e., stroke, chronic ischaemic heart disease) is shown. No patient had a previous history of cardiovascular disease?

  • Table 1 shows one HIV-infected patient diagnosed of “ilicit drug use”. However, in the section “methods”, drug abuse is considered as an exclusion criteria. Could you clarify this issue?

  • Discussion: nucleoside reverse transcriptase inhibitors (NRTIs) are indicated as a risk factor for NAFLD among other liver diseases. However, other HIV-active drugs such as some non- nucleoside reverse transcriptase inhibitors (NNRTIs) exhibit also an increased risk for chronic liver disease (e.g., efavirenz, EFV), while modern NNRTIs (e.g., rilpivirine, RPV) seem to ameliorate liver fibrosis (Benedicto, A. M., Fuster-Martínez, I., Tosca, J., et al. (2021). NNRTI and Liver Damage: Evidence of Their Association and the Mechanisms Involved. Cells, 10(7), 1687. https://doi.org/10.3390/cells10071687; Martí-Rodrigo A, Alegre F, Moragrega ÁB, et al. Rilpivirine attenuates liver fibrosis through selective STAT1-mediated apoptosis in hepatic stellate cells. Gut. 2020 May;69(5):920-932. doi: 10.1136/gutjnl-2019-318372. Epub 2019 Sep 17. PMID: 31530714.). I think the relationship between NNRTI and NAFLD should be adressed in one or two sentences.

Author Response

Manuscript ID: jcm-1784611

Article title: Effects of HIV infection in plasma free fatty acid profiles among people with non-alcoholic fatty liver disease.

The authors would like to thank the Reviewers and Editor for their careful review of our manuscript and their constructive comments and suggestions for improving its quality. The following responses have been prepared to address the referees’ comments point-by-point. Line numbers refer to the revised manuscript with tracked changes.

REVIEWER #2 COMMENTS:

The authors present an interesting paper about differences between HIV infected and HIV uninfected patients regarding the lipid profiles in the setting of non-alcoholic fatty liver disease (NAFLD). The manuscript is well written, and the results and discussion are well explained.

Response: Thank you for the positive assessment of our work.

 However some minor issues should be addressed:

  1. Metabolic syndromeis present in both HIV-infected and uninfected patients, specially among the latter. However, I could not find how metabolic syndrome was defined. Is it defined according to the Executive Summary of the Third Report of the National Cholesterol Education Program (NCEP) (Expert Panel on Detection, Evaluation, and Treatment of High Blood Cholesterol in Adults. Executive Summary of The Third Report of The National Cholesterol Education Program (NCEP) Expert Panel on Detection, Evaluation, And Treatment of High Blood Cholesterol In Adults (Adult Treatment Panel III). JAMA. 2001 May 16;285(19):2486-97. doi: 10.1001/jama.285.19.2486. PMID: 11368702) or another standard?

Response: Thank you for spotting this lack of information in the Methods section. Indeed, we used the NCEP report, which we have referenced in the Methods (lines 84-85, reference 11)

  1. Table 1shows the most common comorbidities within the study population. However, no mention to cardiovascular disease (i.e., stroke, chronic ischaemic heart disease) is shown. No patient had a previous history of cardiovascular disease?

Response: We have added the information on cardiovascular disease (stroke, ischemic heart disease) to Table 1.

  1. Table 1 shows one HIV-infected patient diagnosed of “ilicit drug use”. However, in the section “methods”, drug abuse is considered as an exclusion criteria. Could you clarify this issue?

Response: Thank you for spotting this point that can lead to confusion. There were two patients diagnosed more than 10 years ago with intravenous drug use being the likely route of HIV transmission. However, at present (and for more than 10 years) the participants do not use drugs, so they do not meet an exclusion criterion. We have expanded the eligibility criteria in the Methods section (lines 68-81)

  1. Discussion: nucleoside reverse transcriptase inhibitors (NRTIs) are indicated as a risk factor for NAFLD among other liver diseases. However, other HIV-active drugs such as some non- nucleoside reverse transcriptase inhibitors (NNRTIs) exhibit also an increased risk for chronic liver disease (e.g., efavirenz, EFV), while modern NNRTIs (e.g., rilpivirine, RPV) seem to ameliorate liver fibrosis (Benedicto, A. M., Fuster-Martínez, I., Tosca, J., et al. (2021). NNRTI and Liver Damage: Evidence of Their Association and the Mechanisms Involved. Cells, 10(7), 1687. https://doi.org/10.3390/cells10071687; Martí-Rodrigo A, Alegre F, Moragrega ÁB, et al. Rilpivirine attenuates liver fibrosis through selective STAT1-mediated apoptosis in hepatic stellate cells. Gut. 2020 May;69(5):920-932. doi: 10.1136/gutjnl-2019-318372. Epub 2019 Sep 17. PMID: 31530714.). I think the relationship between NNRTI and NAFLD should be adressed in one or two sentences.

Response:  Thank you for this recommendation. We have expanded the discussion to address this issue (lines 260-262, references 17-18)

Round 2

Reviewer 1 Report

The authors made all the changes requested in the manuscript “Effects of HIV infection on plasma free fatty acid profiles among people with non-alcoholic fatty liver disease”, and was therefore accept in present form.